# Spectral-Kurtosis and Image-Embedding Approach for Target Classification in Micro-Doppler Signatures

Ji-Hyeon Kim, Soon-Young Kwon and Hyoung-Nam Kim *

Department of Electronics Engineering, Pusan National University, Busan 46241, Republic of Korea;
kjihyeon@pusan.ac.kr (J.-H.K.); ysk1680@pusan.ac.kr (S.-Y.K.)
* Correspondence: hnkim@pusan.ac.kr

**Abstract:** Micro-Doppler signature represents the micromotion state of a target, and it is used in target recognition and classification technology. The micro-Doppler frequency appears as a transition of the Doppler frequency due to the rotation and vibration of an object. Thus, tracking and classifying targets with high recognition accuracy is possible. However, it is difficult to distinguish the types of targets when subdividing targets with the same micromotion or classifying different targets with similar velocities. In this study, we address the problem of classification of three different targets with similar speeds and segmentation of the same type of targets. A novel signature extraction procedure is developed to automatically recognize drone, bird, and human targets by exploiting the different micro-Doppler signatures exhibited by each target. The developed algorithm is based on a novel adaptation of the spectral kurtosis technique of the radar echoes reflected by the three target types. Further, image-embedding layers are used to classify the spectral kurtosis of objects with the same micromotion. We apply a ResNet34 deep neural network to micro-Doppler images to analyze its performance in classifying objects performing micro-movements on the collected bistatic radar data. The results demonstrate that the proposed method accurately differentiates the three targets and effectively classifies multiple targets with the same micromotion.

**Keywords:** micro-Doppler; ResNet34; spectral kurtosis; image embedding; target classification

## 1. Introduction

Recently, target detection using commercial broadcast signals and research on passive radar systems have been actively conducted [1–3]. For commercial broadcasting and communication purposes, the signal transmitted from a transmitter can be reflected by hitting a moving target, and a passive radar system estimates the position and speed of the target using the target reflection signal [4].

To increase the accuracy of target detection and tracking, interference signals, such as clutter, should be removed to facilitate the detection of weak target signals [1]. However, existing passive-sensor-based position and speed measurements have limitations in target recognition. To overcome this problem, research on the parameters that represent additional target characteristics is required.

The received echo signal after hitting the target has the Doppler effect, in which the frequency changes according to the relative speed of the moving target. In addition to the distance movement, the target object undergoes micromotions such as vibration and rotation, and the micro-Doppler effect occurs due to these micromotions. This micro-Doppler signal has unique characteristics depending on the target micromotions and has drawn much interest in estimating micromotion parameters and target recognition in the civilian and military fields.

Micro-Doppler can be represented as a micro-Doppler image on the time-frequency axis using short-time Fourier transform (STFT), which confines the radar reflection signal according to time to an arbitrary area, performs Fourier transform and repeats it through

a time delay [5]. The micro-Doppler image is expressed as a sinusoidal wave with the micromotion frequency and initial phase. It can be used to discover unique features of the target, such as the instantaneous motion changes over time of the parts of the target, including motors and wings, and the period of the target's motion. In radar signal analysis, the target's speed change along the radar's line of sight can be represented in a two-dimensional time-frequency domain. This approach involves analyzing micro-Doppler features within this domain and identifying them through their associated frequencies. However, it is difficult to distinguish between the types of targets when subdividing those with the same micromotion or classifying different targets with similar velocities. Therefore, to obtain high classification performance, it is necessary to effectively extract feature vectors based on the micro-Doppler signals of each target and use them for classification.

In this study, to identify objects with different micro-movements, we modeled micro-Doppler signals according to the rotational motion of drone blades, birds' flapping wings, and pedestrians' walking behavior. Micro-Doppler signatures prove instrumental in classifying each target in scenarios where drones and birds exhibit similar flight heights and speeds. Additionally, slow-moving drones or birds may introduce interference signals when the detection target is a person, necessitating their classification. To address this, we extracted the spectral kurtosis from the micro-Doppler images of these three targets as a feature vector. Spectral kurtosis, a statistical feature, effectively reveals non-stationary and/or non-Gaussian behavior in the frequency domain [6]. It assumes very low values in stationary Gaussian signals and dramatically increases when the non-stationary and non-Gaussian behavior of the signal is in the frequency domain. This characteristic has been used to classify payloads of unmanned aerial vehicles (UAV) [7–9].

We used the spectral kurtosis characteristics to obtain the spectral kurtosis values of the micro-Doppler images of three targets: drones, birds, and pedestrians. The kurtosis results indicated that the frequencies with high kurtosis values differed for each target. A signature for classifying the three targets was identified by adopting image embedding in the spectral kurtosis image. This image embedding technique is advantageous in segmenting targets exhibiting identical micromotions and classifying targets moving at similar speeds. The embedding layer facilitates more precise extraction and analysis of signals and objects from the raw feature vectors by converting the high-dimensional input feature vector and sparse data matrix into two dense sub-matrices [10]. Our approach, thus, significantly improves recognition accuracy by refining the identification of unique signatures within the spectral kurtosis images for each target class.

Moreover, we utilized the ResNet34 technique [11,12], a deep-learning algorithm, for classifying the three targets using spectral kurtosis. ResNet34 contains iterative convolutional layers and residual connection structures known as 'shortcuts'. These shortcuts significantly simplify network complexity and enhance the accuracy of recognition tasks. These shortcuts address the vanishing gradient problem, allowing more profound and practical learning. Additionally, comparing the classification performance of the AlexNet [13], VGGNet16 [14], GoogLeNet [15], and ResNet34 algorithms with micro-Doppler images, the ResNet34 algorithm showed the best performance of approximately 90% [16]. This indicates that ResNet34 can efficiently learn complex features and handle feature representations at various levels without being hampered by network depth. Therefore, this attribute of the ResNet34 algorithm is used to distinguish between targets with subtle differences in their micro-Doppler signatures for precise classification. By including ResNet34 in our classification system, we can identify more complex patterns in spectral kurtosis images, thus strengthening the efficacy and reliability of the proposed approach in target classification.

The remainder of this paper is organized as follows. Section 2 describes signal modeling according to the target type. Section 3 introduces spectral kurtosis, a micro-Doppler feature vector, and presents image embedding for effective extraction. The subsequent section describes the application of ResNet, a deep-learning algorithm for target classification. Section 5 analyzes the target classification performance of the ResNet34 deep-learning

algorithm through experiments. Finally, Section 6 concludes the paper with suggestions for future work.

## 2. Micromotion Modeling

The geometry of the rotational motion of the drone blade is illustrated in Figure 1. It is assumed that the distance between the transmitter and the drone blade is equal to that between the receiver and the drone blade. The coordinates of the transmitter are fixed at the origin $(x_0, y_0, z_0 = 0)$, and if the receiver is placed on the same plane separated by $B_L$ on the X-axis, the position of the receiver is $(B_L, 0, 0)$.

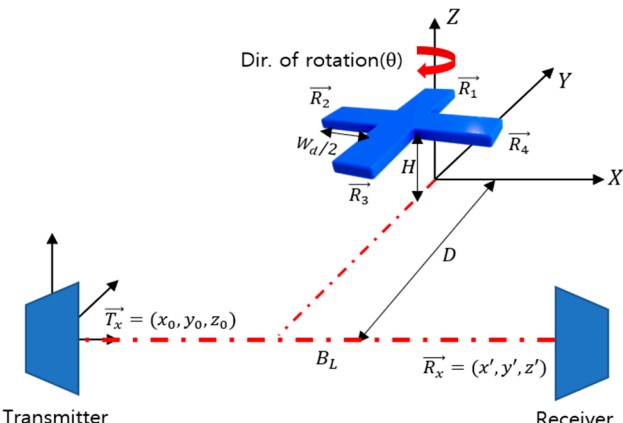

**Figure 1.** Geometry of drone rotor blades.

Four blades rotate around the Z-axis at rotational speed $\theta$. $H$ is the height of the Z-axis of the blade, $D$ is the distance from the transmitter-receiver line to the farthest blade, and $W_d$ is the width of the drone. The position vector of each rotor $\overrightarrow{R_n}$ is defined as follows:

$$\overrightarrow{R_1} = \left( \left( \frac{B_L}{2} \right), D, H \right), \tag{1}$$

$$\overrightarrow{R_2} = \left( \left( \frac{B_L}{2} + \frac{W_d}{2} \right), \left( D - \frac{W_d}{2} \right), H \right), \tag{2}$$

$$\overrightarrow{R_3} = \left( \left( \frac{B_L}{2} \right), (D - W_d), H \right), \tag{3}$$

$$\overrightarrow{R_4} = \left( \left( \frac{B_L}{2} - \frac{W_d}{2} \right), \left( D - \frac{W_d}{2} \right), H \right), \tag{4}$$

Each of these rotors serves as the center of two-blades upon which the blades rotate. Based on the rotor positions in Figure 1, the range profiles of two rotating blades are presented in Figure 2. Each rotor has two blades rotating around a fixed point, creating a bistatic angle with the receiver and transmitter. The figure shows the Doppler effects caused by the blade tip and the blade's range profile with reference to the transmitter and receiver. The range between the transmitter-to-blade-tip is given by $R_{tx\_tip_{nm}}$ and the receiver-blade-tip by $R_{rx\_tip_{nm}}$ for each m-th blade equipped at the n-th rotor (for m = 1, 2 and n = 1, $\cdots$, 4). The angle between the $R_{tx\_tip_{nm}}$ and the $R_{rx\_tip_{nm}}$ is bistatic angle and represented as $\beta_{tip}$ in Figure 2. $\beta_{tip}$ is formed by the blade tip while rotating through 0 to $2\pi$ around the rotor's fixed position. Similarly, $R_{tx\_Fixed_n}$ and $R_{rx\_Fixed_n}$ are the corresponding ranges of a transmitter-to-rotor fixed position and rotor-to-receiver, with bistatic angle $\beta$. The scattered signal causes an additional modulation due to the rotating blades, and thus the angular frequency $\omega_{sc}$ is the combination of the Doppler component $f_d$ due to linear motion and the micro-Doppler $f_{md}$ due to the blade rotation. Therefore, as

described in [17], we can obtain $\omega_{sc} = 2\pi f_d t + 2\pi f_{md} t$. Here, the general Doppler equation for the bistatic arrangement is given by

$$f_d = \frac{2V}{\lambda} \cos\left(\frac{\beta}{2}\right) \cos(\delta),$$
$$\delta = \pi - \beta/2 \tag{5}$$

where $V$ is the velocity of the drone, $\left[\cos\left(\frac{\beta}{2}\right)\cos(\delta)\right]$ is the coefficient term of the bistatic geometry, and $\lambda$ is the wavelength of the scattered signal.

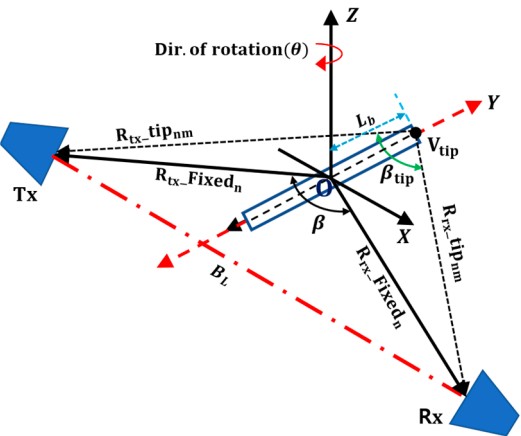

**Figure 2.** The geometry of drone rotating blades range profiles.

The micro-Doppler equation $f_{md}$ is obtained by rotating blade. The blade's yaw angle variation is determined by the rotation rate, the blade length $L_b$, and the initial angle given by $\theta_0$. When the blade rotates at a constant rotation rate $\Omega$ over time $t$, a yaw angle is formed by $(\Omega t + \theta_0)$ and the velocity is caused by the blade tip. The velocity of the blade tip $V_{\text{tip}}$ is a function of the Doppler effect and is expressed by

$$V_{\text{tip}} = 2\pi L_b \cos(\Omega t + \theta_0). \tag{6}$$

In this case, the phase function of the blade can be obtained by

$$\Phi(t) = -\frac{2\pi}{\lambda} \frac{L_b}{2} \cos\left(\frac{\beta}{2}\right) \cos(\delta) \cos(\Omega t + \theta_0). \tag{7}$$

A detailed derivation of the Doppler equation and phase function is in [17,18].

As shown in Figure 2, the initial angle $\theta_0$ is assumed to lie at the origin, and this made the initial angle to be zero, leaving only the blade rotation rate $\Omega t$. We use the Bessel function of the first kind to obtain the micro-Doppler frequency of each blade. This is achieved by differentiating phase function $\Phi(t)$ for time $t$.

$$f_{md} = \frac{1}{2\pi} \frac{d}{dt} \Phi(t) \tag{8}$$

$$= \frac{1}{2\pi} \frac{d}{dt}\left(-\frac{2\pi}{\lambda} \frac{L_b}{2} \cos\left(\frac{\beta}{2}\right) \cos(\delta) \cos(\Omega t + \theta_0)\right) \tag{9}$$

$$f_{md\_max} = \frac{V_{\text{tip}}}{\lambda} \cos\left(\frac{\beta}{2}\right) \cos(\delta) \tag{10}$$

Based on the described geometry of Figures 1 and 2, the bistatic angle made by the blade tip $\beta_{\text{tip}}$ is used in the Doppler equation; hence, the micro-Doppler equation is obtained by $f_{md} = \frac{V_{\text{tip}}}{\lambda} \cos\left(\frac{\beta_{\text{tip}}}{2}\right) \cos(\delta)$.

The wing motion model of a bird is shown in Figure 3. With the origin of the XYZ-plane as the body of the bird, the two blue circles represent the scattering points caused by micro-movements when the bird flaps its wings. The upper part of the wing (upper arm) has a length of $L_1$, and the lower part of the wing (lower arm) has a length of $L_2$; the wing flapping has a motion frequency $f_{flap}$. The movement periods of birds and their micro-movements over time can be expressed as follows [5]:

$$
\begin{pmatrix} \psi_1 \\ x_1 \\ y_1 \\ z_1 \end{pmatrix}(t) = \begin{pmatrix} 40\cos\left(2\pi f_{flap}t\right) + 15 \\ 0 \\ L_1\cos\psi_1(t) \\ y_1(t)\tan\psi_1(t) \end{pmatrix},
\tag{11}
$$

$$
\begin{pmatrix} \psi_2 \\ \phi_2 \\ x_2 \\ y_2 \\ z_2 \end{pmatrix}(t) = \begin{pmatrix} 30\cos\left(2\pi f_{flap}t\right) + 40 \\ 20\cos\left(2\pi f_{flap}t\right) \\ -(y_2(t) - y_1(t))\cdot\tan(\phi_2(t)/\cos(\psi_1(t) - \psi_2(t))) \\ L_1\cos\psi_1(t) + L_2\cos\phi_2(t)\cdot\cos(\psi_1(t) - \psi_2(t)) \\ z_1(t) + (y_2(t) - y_1(t))\cdot\tan(\psi_1(t) - \psi_2(t)) \end{pmatrix}
\tag{12}
$$

where $\psi_1$ is the angle of the upper arm, $\psi_2$ is the angle of the lower arm with respect to the upper arm, and $\phi_2$ is the angle of change in the axis of the lower arm.

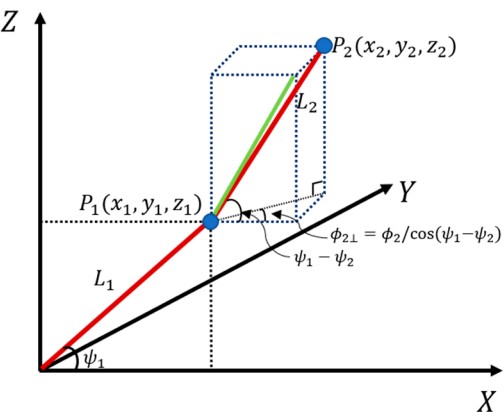

**Figure 3.** Kinematic model of a flying bird.

Figure 4 shows the micromotion model of a pedestrian when the XYZ-plane's origin is centered on the human spine and pelvis [12]. The model simulates the micro-movement of body parts during human walking; it is divided into the head, shoulders, elbows, knees, ankles, and other body parts, marked by a circle. When a person walks at speed of $V_R$, the length of the walking cycle $R_c$ and cycle duration $T_c$ can be obtained as follows:

$$
R_c = 1.346 \times \sqrt{V_R},
\tag{13}
$$

$$
T_c = R_c / V_R.
\tag{14}
$$

When a pedestrian walks, the change in the micromotion $Tr_{F/B}$ due to the backward and forward movements of the legs can be expressed by Equation (15). In Equation (16), $a_{F/B}$ is the angular velocity for the back-and-forth motion, $\phi_{F/B}$ of Equation (17) is the change angle with respect to the X-axis, and $t_R$ of Equation (18) is the time normalized to the duration of one step.

$$
Tr_{F/B} = a_{F/B}\sin(2\pi(2t_R + 2\phi_{F/B}))
\tag{15}
$$

$$a_{F/B} = \begin{cases} -0.084V_R(V_R - 1) & (V_R < 0.5) \\ -0.021 & (V_R > 0.5) \end{cases} \tag{16}$$

$$\phi_{F/B} = 0.768 - 0.752T_C \tag{17}$$

$$t_R = t/T_C \tag{18}$$

In addition, the forward/backward/left/right/torsion rotational movements of the pelvis that occur to make the back-and-forth movement of the leg are shown in Equations (19)–(21). Equations (22)–(24) represent the angular velocity of the forward/backward/left/right/torsion rotational movements of the parts constituting the target, such as the motors and wings.

$$Ro_{F/B} = -ar_{F/B}(1 - \sin(2\pi(2t_R - 0.1))) \tag{19}$$

$$Ro_{L/R} = \begin{cases} -ar_{\frac{L}{R}}\left(1 - \cos\left(2\pi\left(\frac{10t_R}{3}\right)\right)\right) & (0 \le t_R < 0.15) \\ -ar_{\frac{L}{R}}\left(1 + \cos\left(2\pi\left(\frac{10(t_R - 0.15)}{7}\right)\right)\right) & (0.15 \le t_R < 0.5) \\ -ar_{\frac{L}{R}}\left(1 + \cos\left(2\pi\left(\frac{10(t_R - 0.5)}{3}\right)\right)\right) & (0.5 \le t_R < 0.65) \\ -ar_{\frac{L}{R}}\left(1 - \cos\left(2\pi\left(\frac{10(t_R - 0.65)}{7}\right)\right)\right) & (0.65 \le t_R < 1) \end{cases} \tag{20}$$

$$Ro_{Tor} = -ar_{Tor}\cos(2\pi t_R) \tag{21}$$

$$ar_{F/B} = \begin{cases} -8V_R(V_R - 1) & (V_R < 0.5) \\ 2 & (V_R > 0.5) \end{cases} \tag{22}$$

$$ar_{L/R} = 1.66V_R \tag{23}$$

$$ar_{Tor} = 4V_R \tag{24}$$

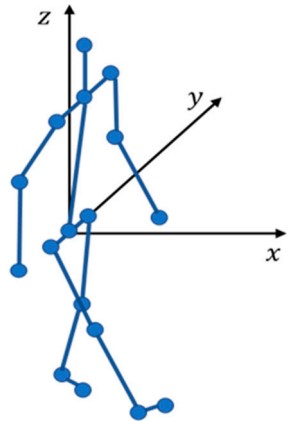

**Figure 4.** Kinematic model of a walking human.

## 3. Spectral Kurtosis-Based Target Classification

### 3.1. Spectral Kurtosis

The targets such as drones, birds, and pedestrians have different micromotions, they can be expected to have differing micro-Doppler characteristics depending on the target type. The micro-Doppler attributes of the three targets are obtained using the spectral kurtosis of the micro-Doppler image obtained using the signals theoretically modeled for the micromotion of the three targets described in Section 2.

Based on the collected reflection signal, a micro-Doppler image is obtained by applying an STFT:

$$\text{STFT}(\nu, k) = \sum_{n=0}^{N-1} s(n)w^*(n-k)e^{-\frac{j2\pi\nu n}{N}}, \quad k = 0, \cdots, K-1 \tag{25}$$

where $w(\cdot)$ is the smoothing window function with length $K$ used in the STFT, $N$ is the length of the signal, and it is assumed that $K$ is smaller than $N$. $\nu \in [-1/2, 1/2]$ represents the normalized frequency. Specifically, the time window length $K$ is chosen based on the target's speed. In the case of drones [7,8], the STFT window length is determined by dividing the total signal sample size by a factor of 16. Under the assumption that the movement speeds of birds and pedestrians are comparable to those of drones, we apply the same factor.

Spectral kurtosis can extract and observe additional information regarding non-Gaussian signals that would not be revealed when computing the classic power spectral density. Additionally, computing the kurtosis for each frequency contained in the signal emphasizes the non-stationarity of the signal and allows the location of signal transients in the frequency domain. For Gaussian and stationary signals, the spectral kurtosis assumes low values, whereas higher values correspond to non-stationarity. Therefore, spectral kurtosis is a statistical tool [6] used with power spectral density to identify nonstationary and/or non-Gaussian frequency components [7].

A simple definition of the spectral kurtosis is provided using the normalized fourth-order moment of the STFT magnitude. Spectral kurtosis $\Psi(\nu)$ is proportional to the ratio between the fourth-order moment of its STFT and the square modulus of the second-order moment of its STFT, namely

$$\Psi(\nu) = \frac{\frac{1}{K}\sum_{k=0}^{K-1}|\mathrm{STFT}(\nu,k)|^4}{\frac{1}{K}\left(\sum_{k=0}^{K-1}|\mathrm{STFT}(\nu,k)|^2\right)^2} - 2. \tag{26}$$

Figure 5 shows the spectral kurtosis of the micro-Doppler images according to the rotation of drone blades, bird wings flapping, and pedestrians' walking behavior. The frequencies with high kurtosis values and amplitudes differ for each target from the spectral kurtosis results for the three targets. However, the differences may be indistinguishable for precise classification; thus, additional steps are required to classify the three targets reliably.

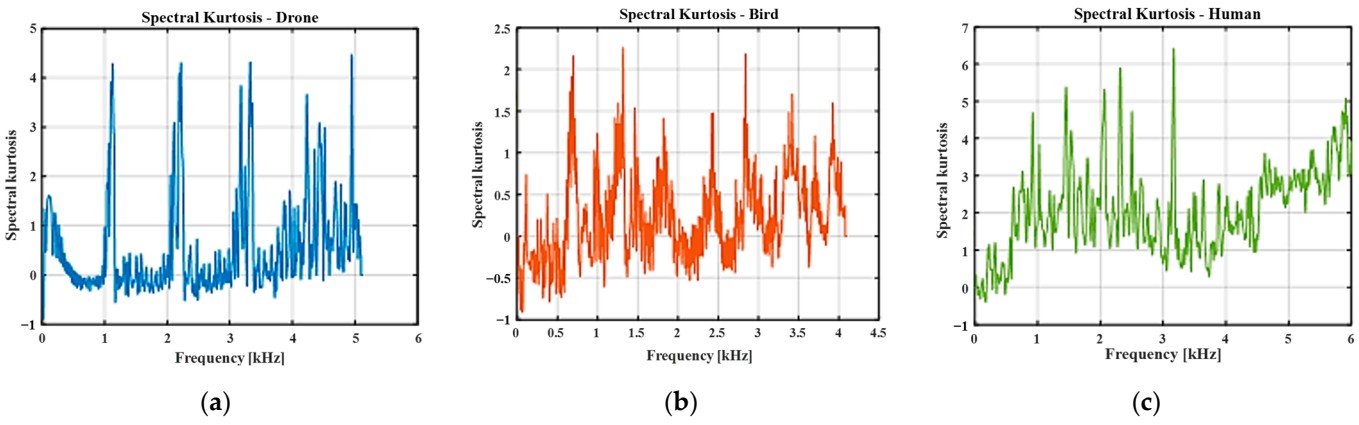

(a)  (b)  (c)

**Figure 5.** Spectral kurtosis of targets (simulation data): (**a**) drone, (**b**) bird, (**c**) human.

In this study, an embedding function was applied to the obtained spectral kurtosis images to improve the classification performance for each target. The details of the embedding function are presented in the following section.

### 3.2. Feature Extraction Using an Embedding Function

Targets with different micromotions can be accurately classified using the spectral kurtosis feature. However, classifying targets is challenging if they are of the same type and differ in speed or size. To overcome this difficulty, we added an embedding layer to the kurtosis feature image to distinguish targets with the same micromotion into detailed categories.

Different embedding layers of each high-dimensional input feature vector and sparse data matrix in two dense low-rank matrices can lead to good recognition accuracy. To obtain the signal $u$ and the target $v$ of the primitive feature vectors in the embedding space, the following steps are performed: Let $P \in \mathbb{R}^{m \times k}$ represent the signal embedding matrix and $Q \in \mathbb{R}^{n \times k}$ represent the target embedding matrix, where $k$ represents the embedding dimension. The signal latent vector $p_i$ and target latent vector $q_j$ are then computed as in [10].

$$p_i = P^T u_i \tag{27}$$

$$q_j = Q^T v_j \tag{28}$$

To illustrate the concepts discussed above, the rating matrix $R$ is first decomposed into two low-rank matrices, $P$ (the signal latent matrix) and $Q$ (the target latent matrix) satisfying $R \approx PQ^T$. To find the matrices, we used an embedding loss function that minimizes the difference between the actual and approximate values defined by [18].

$$L_{EM} = \min_{P,Q} \left[ \sum_{(i,j) \in R_{known}} \left( r_{ij} - p_i q_j^T \right)^2 \right] \tag{29}$$

The feature vector in the embedding space is obtained and thus each target could be classified through the embedding layer using the spectral kurtosis image obtained according to each micromotion as an input.

## 4. Deep Neural Network Algorithm

Artificial neural networks are widely used in various fields because of their excellent performance in image classification and recognition [11]. A convolutional neural network is a multilayer feedforward neural network with a complicated structure, good fault tolerance, and self-learning capabilities. It addresses problems in complex environments and with unclear backgrounds. ResNet [11,12] is the most commonly used among the several convolutional neural network algorithms because of its excellent classification performance. It has a structure in which $3 \times 3$ convolutional layers are iterated.

Five versions of ResNet are recognized, depending on the number of layers: ResNet-18, ResNet-34, ResNet-50, ResNet-101, and ResNet-152. As the number of layers increases, the amount of calculation and the number of parameters increases, but the performance can be enhanced in terms of accuracy or test error. The top five validation errors of a single model result on the ImageNet set were approximately 5.6% for ResNet34, 5.25% for ResNet50, 4.6% for ResNet101, and 4.49% for ResNet152 [11]. When the image classification performances of ResNet34 and ResNet101 algorithms are compared, the error rate of the ResNet101 algorithm is better than 1%. Regarding computation, the ResNet34 algorithm has floating point operations per second of $3.6 \times 10^9$, and the ResNet101 algorithm has more than double, $7.6 \times 10^9$ [11]. Therefore, the ResNet34 algorithm provides a good trade-off between performance and complexity.

Figure 6 shows the shortcut structure of the network used in the ResNet algorithm. This structure is pivotal to the algorithm's function: In the ResNet34 algorithm, after the input data traverses 16 shortcut structures, the algorithm can accurately determine the data's classification. Each shortcut, characterized by identity mappings, effectively passes the input value directly without modifying it through convolutional layers. This unique approach enhances the algorithm's ability to preserve essential information from the input while processing it through the network, ultimately leading to the final classification of the input data. The weight layer in the network is responsible for convolution, a critical feature extraction and learning process. The activation function used is the rectified linear unit (ReLU), which is crucial in preventing performance degradation, a common issue in deep networks. $H(x)$ is the actual expected output of a particular layer, and $x$ is the input. Traditional neural networks aim to approximate $H(x)$ directly, but ResNet learns in the direction where $H(x)$ becomes 0. This is achieved by learning the residual,

$F(x) = H(x) - x$. The unique approach, termed residual learning, simplifies optimizing the network's parameters and addresses problems like vanishing and exploding gradients. The identity mappings in shortcut connections can improve the recognition accuracy while reducing the network complexity.

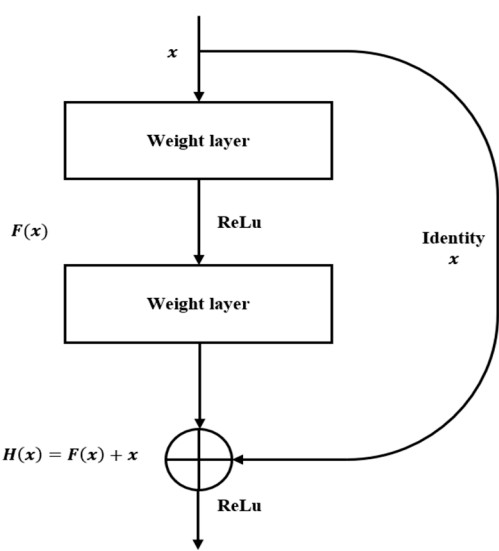

**Figure 6.** "Basic-Block" building block for ResNet.

## 5. Classification Experiments

The following section shows the performance of the proposed target classification algorithm for drones, birds, and pedestrians using spectral kurtosis, using the ResNet34 as described in Section 4.

### 5.1. Experimental Environments

The drone, bird, and human datasets used for classification are measurement signal data collected in various signal-to-noise-ratio environments for target detection, identification, and tracking. These datasets are publicly available data provided by [19–21]. Tables 1 and 2 present the types and details of the drones and bird targets. For the pedestrian target, data from subjects with various characteristics, such as age, gender, and height, were used. Of these datasets, 70% were used as training data and 30% as test data. Further, the Adam optimizer [11] was used for the weight-updating process: the number of epochs was 100, and the mini-batch size was 64.

**Table 1.** Details of the drone databases.

| Drone Type | Dimension (cm) | Range (m) |
|---|---|---|
| Bepop | 38 × 33 × 3.6 | 50 |
| AR | 61 × 61 × 12.7 | 50 |

**Table 2.** Details of the bird flapping databases.

| Bird Type | Flap Angle (rad) | Velocity (m/s) |
|---|---|---|
| Chukar | 2.5 | 1.2 |
| Pigeon | 1.57 | 1.5 |

### 5.2. Performance Evaluation Metrics

A confusion matrix is a $N \times N$ matrix used to evaluate the performance of a classification model, where $N$ represents the number of target classes. This matrix compares the actual target value with the value predicted by the machine learning model. This provides

a visual indicator of the accuracy and error of the classification model. Figure 7 shows a confusion matrix for multi-class model classification with $N$ classes. The results for good/incorrect classification for each class are collected as a confusion matrix $C := (c_{ij})$, where $c_{ij}$ is the number of data points, where actual class $i$ is estimated by class $j$. The actual class is the actual value, the estimated class is the expected value, TP and TN are the parts where the actual value is correctly predicted, and FP and FN indicate the parts where the actual and expected values differ. Each term is explained as follows:

1. TP (true positive): accurate classification of the category of interest
2. TN (true negative): accurate classification of items not in the category of interest
3. FP (false positive): misclassification of noninterest categories as interest categories
4. FN (false negative): incorrect classification of a category of interest as not a category of interest.

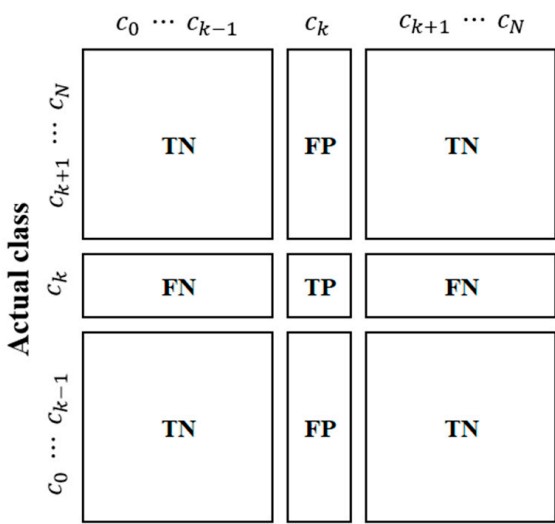

**Figure 7.** Confusion matrix for multi-class classification.

Based on these four pieces of information, three measures can be determined: accuracy, precision, and recall. Accuracy is the percentage of correctly classified data out of the total data, indicating the accuracy of the model classification. The precision indicates the number of correctly predicted cases that correctly classify the category of interest. This metric allowed the determination of the reliability of the model. Recall is a measure compared to precision. It refers to the number of data predicted for a category of interest using a model from the actual number of data. The metrics can be calculated as follows:

$$\text{Accuracy} = \frac{\text{TP} + \text{TN}}{\text{TP} + \text{FP} + \text{TN} + \text{FN}}, \tag{30}$$

$$\text{Precision} = \frac{\text{TP}}{\text{TP} + \text{FP}}, \tag{31}$$

$$\text{Recall} = \frac{\text{TP}}{\text{TP} + \text{FN}}. \tag{32}$$

The F1 score is a performance metric used in machine learning to evaluate the predictive ability of an AI model concerning each class. It is the harmonic mean of precision and recall, effectively balancing the two when both are considered equally important. The F1 score ranges from 0.0 to 1.0, with a score closer to 1.0 indicating better model performance. While a model with an F1 score of 0.7 or higher is often considered suitable the threshold for a 'suitable' score can vary widely based on the specific application and

requirements. In contexts where the cost of false positives and false negatives is high, a higher F1 score would be necessary.

$$\text{F1 score} = 2 \times \frac{\text{Precision} \times \text{Recall}}{\text{Precision} + \text{Recall}} \tag{33}$$

*5.3. Classification Experiments*

5.3.1. Experiment 1: Three-Class Classification of Different Micromotion Targets

This subsection presents the classification capabilities of the algorithm discussed in Section 4, which is designed to classify different targets based on their unique micro-Doppler signatures. To analyze the performance of the proposed technique, we augmented the ResNet34 algorithm with an image-embedding layer that processes the spectral kurtosis calculated using Equation (26) for drone, bird, and pedestrian targets.

Figure 8 shows the spectral kurtosis results of the target as a drone, bird, and human measurement data for classification. Similar to the spectral kurtosis of the simulation data in Figure 5, the spectral kurtosis results for the three subjects can be seen to be different for each subject. However, since the simulation data shown in Figure 5 are modeled signals in an environment where an ideal bistatic configuration is assumed, the difference from the results in Figure 8 is expected to be caused by bistatic distortion due to the bistatic radar configuration.

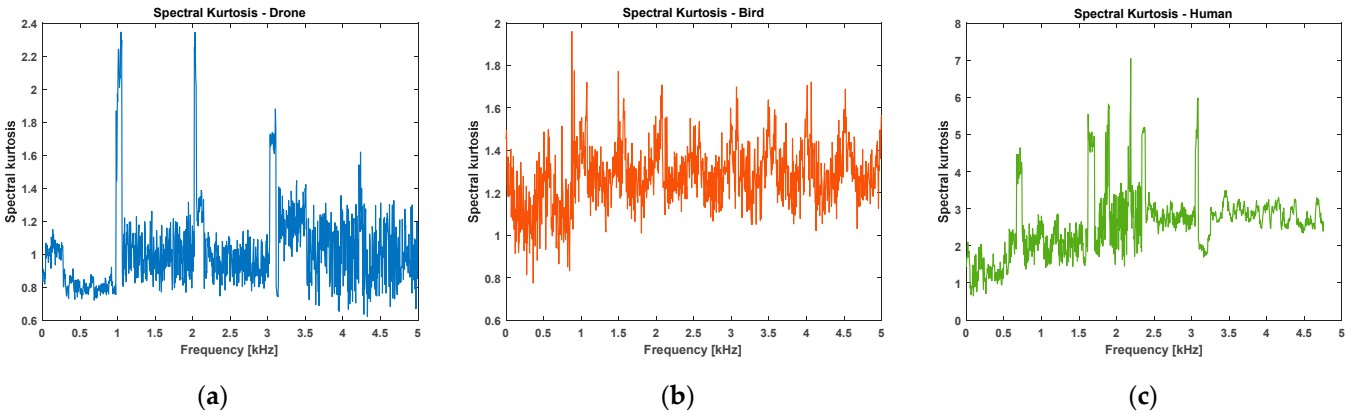

(**a**)  (**b**)  (**c**)

**Figure 8.** Spectral kurtosis of targets (measurement data): (**a**) drone, (**b**) bird, (**c**) human.

Figure 9 shows the classification results in a confusion matrix, where the actual and predicted class matches for each target type are indicated in red, and the misclassifications are shown in blue. This visualization confirms that the algorithm achieves a high classification accuracy, averaging 98% across all target types. In contrast, instances of incorrect classification are rare, representing approximately 1.5% of the cases.

|     | Dr    | Br    | Hm   |
|-----|-------|-------|------|
| **Dr** | 98.19 | 1.81  | 0    |
| **Br** | 1.68  | 97.28 | 1.04 |
| **Hm** | 0     | 0     | 100  |

**Figure 9.** Confusion matrix of Experiment 1.

Figure 10 shows the accuracy, precision, and recall, which are the measures of the algorithm performance calculated using Equations (30)–(32), respectively. In the performance graph, blue represents accuracy, orange indicates precision, and gray corresponds to recall. With accuracy and precision exceeding 98% for each of the three targets and a recall above 97%, the ResNet model of the proposed method demonstrates commendable classification performance.

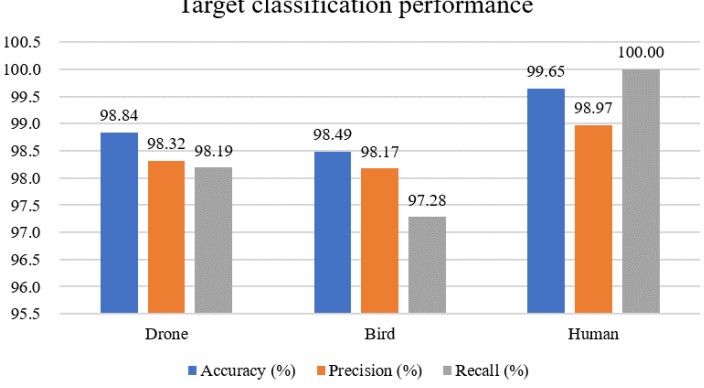

**Figure 10.** Target classification performance of Experiment 1.

The performance of the target classification for drones, birds, and humans using the F1 score metric using a bar graph is shown in Figure 11. The model exhibits high F1 scores across all categories. Specifically, humans have an F1 score of approximately 0.9948, drones register around 0.9825, and birds score about 0.9772. Each score is notably high, underscoring the model's effectiveness and reliability in differentiating these targets.

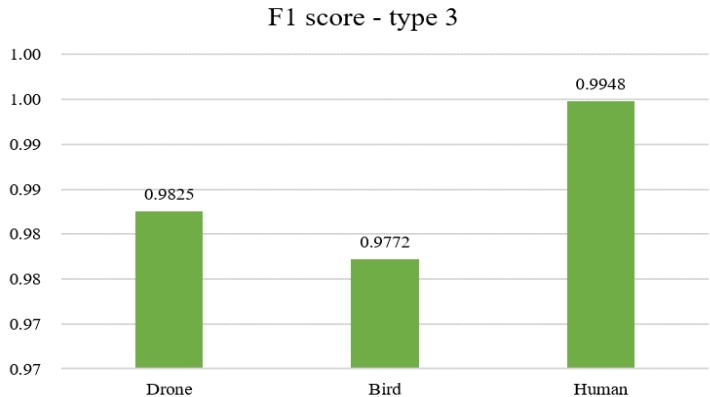

**Figure 11.** F1 score graph of Experiment 1.

Additionally, we calculated four indices to compare the classification performance of the existing and proposed techniques and show them in Table 3. The existing technique adopted spectral kurtosis with micro-Doppler images but does not apply an embedding layer. The spectral kurtosis-based methods, encompassing both approaches with and without the embedding layer, outperform the micro-Doppler image-based technique across all four metrics. Spectral kurtosis requires more straightforward input data than conventional deep-learning models that rely on micro-Doppler imagery. This distinction enhances the classification of the three entities by filtering out shared features in micro-Doppler images.

A specific observation is that traditional micro-Doppler imaging techniques achieved a classification accuracy of only 85% for drones and birds. As both drones and birds are inherently similar in size and movement patterns, it is challenging to distinguish using the micro-Doppler image. However, the spectral kurtosis technique performed well for both objectives, achieving an accuracy of 94%. In particular, the proposed ResNet algorithm is integrated with the embedding layer and provides excellent classification results for

pedestrians, drones, and birds. Considering the F1 score, which is a crucial indicator of model reliability, the score of the existing algorithm is around 0.9. On the other hand, the proposed model incorporating the embedding layer boasts a score of 0.98. These results demonstrate its superiority as a near-optimal target classification algorithm.

**Table 3.** Performance of the classification methods.

| Classification Methods | | Drones | Birds | Human |
|---|---|---|---|---|
| ResNet using micro-Doppler image [16] | Accuracy (%) | 90.9 | 90.9 | 93.7 |
| | Precision (%) | 85.3 | 87.7 | 90.9 |
| | Recall (%) | 87.8 | 84.6 | 90.9 |
| | F1-score | 0.8653 | 0.8612 | 0.909 |
| ResNet using spectral kurtosis (w/o Embedding) [12] | Accuracy (%) | 95.7 | 95.8 | 97.9 |
| | Precision (%) | 93.5 | 93.8 | 96.8 |
| | Recall (%) | 93.5 | 93.5 | 97.2 |
| | F1-score | 0.935 | 0.9365 | 0.97 |
| ResNet using spectral kurtosis (with Embedding) | Accuracy (%) | 98.8 | 98.5 | 99.7 |
| | Precision (%) | 98.3 | 98.2 | 99.0 |
| | Recall (%) | 98.19 | 97.28 | 100 |
| | F1-score | 0.9825 | 0.9772 | 0.9948 |

5.3.2. Experiment 2: Five-Class Classification of Different and Same Micromotion Targets

In this section, we evaluate the performance of the classification algorithm with the same micromotion into four classes: two types of drones and two types of birds (Tables 1 and 2). The experiment result shows that the proposed method classified five-class targets correctly as drone 1, drone 2, bird 1, bird 2, and pedestrian.

Figure 12 shows the confusion matrix for target classification. The classification performance between the actual class of each subject, matching with the predicted class, averaged 93%. When the actual and expected classes did not match, the probability of incorrectly predicting drones 1 and 2 was 6.27% and 3.89%, respectively, more significant than the probability of misclassifying micro-movements as other birds or pedestrians. Nevertheless, the probability of adequately classifying each drone is approximately 94%. However, the classification performance for birds was slightly degraded. Although the two types of birds have different flight heights and wing flapping angles, their body sizes and flight speeds are similar, resulting in a probability of misclassification higher than that of drones or pedestrians. In the case of pedestrians, compared with drones and bird targets, there are many micro-Doppler signals according to body movements, resulting in distinct features and better classification performance using deep learning.

| | Dr1 | Dr2 | Br1 | Br2 | Hm |
|---|---|---|---|---|---|
| **Dr1** | 93.46 | 6.27 | 0.27 | 0 | 0 |
| **Dr2** | 3.89 | 94.57 | 0.18 | 1.36 | 0 |
| **Br1** | 0.84 | 0.05 | 87.62 | 10.57 | 0.92 |
| **Br2** | 0.15 | 0.64 | 10.34 | 88.75 | 0.12 |
| **Hm** | 0 | 0 | 0 | 0 | 100 |

**Figure 12.** Confusion matrix of Experiment 2.

Figure 13 illustrates the measurement parameters of the proposed algorithm, such as accuracy, precision, and recall, using the confusion matrix. The classification accuracy across the five target classes exhibits good performance, averaging 97%. The orange and grey graphs symbolize the precision and recall metrics, respectively, which are pivotal indicators of the algorithm's reliability.

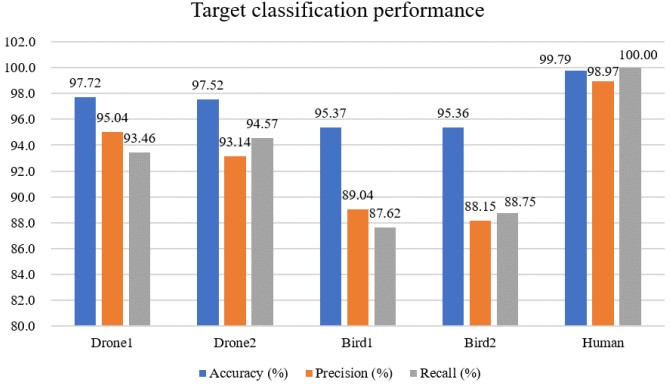

**Figure 13.** Target classification performance of Experiment 2.

When differentiating between two drones, two types of birds, and a pedestrian, the algorithm achieved an average performance rate of 92%. The precision rate of the bird classification is close to 90%, but it was marginally lower than the performance rates of the other target classifications. Remarkably, the classification performance for pedestrians stood out as near-perfect. This result is assigned to the distinct micro-movements of pedestrians, which are significantly differentiated from the micromotions of drones or birds.

Figure 14 shows the F1 scores for the five classification type categories. In particular, the human category achieved an almost perfect F1 score of 0.9948. The drone categories, Drone1 and Drone2, both recorded high scores of about 0.94. Conversely, Bird1 and Bird2 score around 0.88, slightly lower than the drone and human categories but highlighting the model's reliability within the classification spectrum.

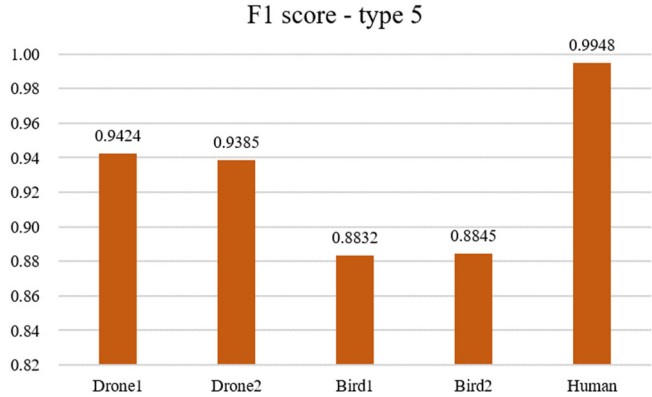

**Figure 14.** F1 score graph of Experiment 2.

5.3.3. Experiment 3: Three-Class Target Classification under Different SNR Scenarios

In the corresponding section, we analyze the performance of a three-class target classification under various SNR scenarios. A significant challenge encountered during Experiments 1 and 2 was the inability to accurately determine the SNR values of the measurement data, rendering a direct performance assessment impractical under a range of SNR conditions. Therefore, we utilized artificially simulated data, as theoretically modeled in Section 2, to effectively evaluate the algorithm's performance across different SNR levels. We set the SNR range for the generated data from −20 dB to 10 dB and derived four performance metrics at 5 dB intervals.

Figure 15 presents the performance metrics of a classification algorithm for three different target classes across varying SNR levels. The analysis shows enhanced accuracy, precision, recall, and F1 scores as SNR increases. Human targets achieve high accuracy even at lower SNRs, indicating the algorithm's proficiency in identifying human signatures. Drone classification remains stable across SNRs, suggesting distinctive drone signatures, while bird detection improves significantly with higher SNR. The F1 score, a metric combining precision and recall, shows uniform improvement across all classes, and human target classification performs better over the entire SNR range than the other two targets. The algorithm thus demonstrates robustness across varying signal qualities.

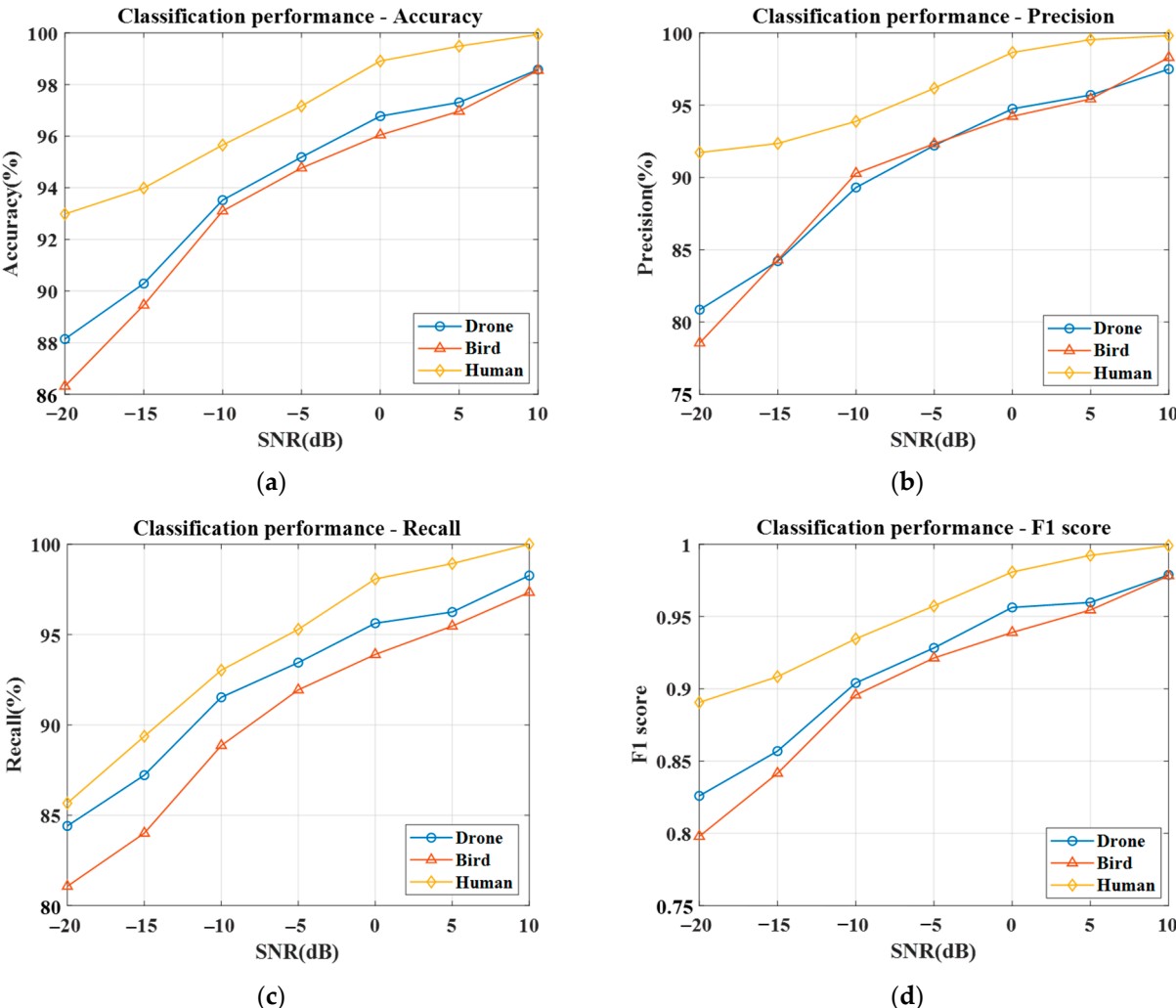

**Figure 15.** Target classification performance of Experiment 3: (**a**) accuracy, (**b**) precision, (**c**) recall, (**d**) F1 score.

Throughout these experiments, we have demonstrated the robustness and effectiveness of the proposed algorithm for classifying targets based on their micro-Doppler signatures. Including an embedding layer, notably when processing spectral kurtosis, has markedly improved the classification performance across different and similar micro-motion targets. The results reiterate the importance of considering the inherent micro-Doppler characteristics of various entities and the potential of deep learning in enhancing target classification.

## 6. Conclusions

In this study, we addressed the problem of classifying three targets with similar speeds and segmenting targets of the same type. After extracting the spectral kurtosis as a micro-Doppler feature, a deep learning technique was used to classify the three targets. In addition, we proposed an effective classification system that can classify the spectral kurtosis of objects with the same micromotion by adding an image-embedding layer. Here, we simulated three-class classifications with targets performing different micromotions and five-class classifications with two types of targets with the same micromotion. We analyzed the classification performance of each target using the ResNet34 deep-learning algorithm. It has an average performance of over 97% in terms of accuracy, precision, and reproducibility. The F1 score was 0.93, which shows that it outperforms the existing classification techniques. To enhance the rigor and completeness of our study, we also included the classification algorithm's performance across different SNR scenarios across various SNR scenarios. The performance results show the algorithm's robustness and dependability over a spectrum of signal qualities. In the future, we will try to train and test our method on a much broader class of three targets. We will consider jointly using estimates of object speed and median radar cross-section with the micro-Doppler data to improve classification performance. Additionally, we plan to study the parameter settings by considering the computational cost of the target classification technique and quantify accuracy as a function of data rate and the geometrical diversity of sensors. By applying this method to the field of target detection, it is expected that higher accuracy can be obtained by estimating the position and speed of the target of interest.

**Author Contributions:** Conceptualization, J.-H.K.; methodology, J.-H.K.; software, J.-H.K.; formal analysis, J.-H.K.; investigation, J.-H.K. and S.-Y.K.; writing—original draft preparation, J.-H.K.; writing—review and editing, J.-H.K., S.-Y.K. and H.-N.K.; supervision, H.-N.K.; project administration, H.-N.K. All authors have read and agreed to the published version of the manuscript.

**Funding:** This work was supported by the National Research Foundation of Korea (NRF) grant funded by the Korea government (MSIT) (No. 2021R1F1A1060025).

**Data Availability Statement:** This work used a publicly available drone dataset from Qatar University, Qatar. Additionally, the following information was provided regarding the availability of bird wing beat data: Dryad: 10.5061/dryad.1f5h4. Human raw Doppler data was collected at the Università Politecnica delle Marche, Ancona, Italy.

**Conflicts of Interest:** The authors declare no conflicts of interest.

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
