# Peer review of "Spectral-Kurtosis and Image-Embedding Approach for Target Classification in Micro-Doppler Signatures"

_electronics, doi:10.3390/electronics13020376_

Round 1

Reviewer 1 Report

Comments and Suggestions for Authors

1. The author should clarify which scenarios may face classification issues for birds, drones, and humans.

2. Scattering points should be used for modeling the three types of targets, and the distribution of scattering points should be explained.

3. The article uses Spectral Kurtosis to classify the simulated target echoes. Can the statistical distribution of the measured data for the three types of targets be provided, and whether the conclusions are consistent with the simulation data.

Comments on the Quality of English Language

Minor editing of English language required

Author Response

We sincerely appreciate the thorough review of our paper. Your valuable insights and feedback have been carefully considered, and we have made necessary revisions to improve the manuscript. The revisions of the issues are shaded by yellow color and English revision is shaded by green color. Your contributions have greatly benefited our paper presentation.

Reviewer 2 Report

Comments and Suggestions for Authors

This paper proposes a spectral-kurtosis and image-embedding approach for target classification in micro-doppler signatures to address the problem of classifying three different targets with similar speeds and segmentation of the same type of targets. Specific comments and suggestions are as follows.

1. How does Eq. (5) been obtained? Please give the derivation process in the revised manuscript.

2.STFT is one kind of time-frequency tool, the time window is important for the performance of STFT, please give the proper criteria for window selection.

3. The bistatic distortion introduced by the bistatic observation configuration is inevitable, but it is not described in the paper. Please give the explanation.

4For the rigor and completeness of the paper, please supplement the performance analysis under different SNR scenarios in the experiment.

5The grammar and spelling errors should be checked carefully in this paper.

Comments on the Quality of English Language

The grammar and spelling errors should be checked carefully in this paper.

Author Response

We would like to thank the reviewer for the thorough and comprehensive review of our paper. We have carefully read all your comments and made corresponding modifications to our revision. English revision is shaded by green color and the revisions of the issues raised by reviewer are shaded by yellow color. The presentation of the paper has benefited greatly thanks to your contribution. 

Reviewer 3 Report

Comments and Suggestions for Authors

1.  For future research, the authors should test and train their algorithm on a much wider class of birds, drones and humans.

2.  For future research, the authors should attempt to invent an algorithm that quantifies the uncertainty of the decisions, rather than merely announcing a decision.

3. For future research, the authors should consider using estimates of object speed and median radar cross-section (and fluctuation statistics of speed & RCS) jointly with the micro-Doppler data.  Intuitively, this should improve performance, but if not, it would be interesting to quantify this.

4. For future research, the authors should attempt to compute a theoretical bound on accuracy, perhaps using tools like the Chernoff bound.

5. For future research, the authors should quantify accuracy as a function of signal-to-noise ratio, data rate and geometrical diversity of sensors.

Author Response

Thank you for your valuable comments. The revisions of the issues raised by reviewer are shaded by yellow color and English revision is shaded by green color. Through your review, we believe this article will be able to present research results in Electronics. Furthermore, your insights and suggestions for future research directions will significantly guide our studies

Round 2

Reviewer 1 Report

Comments and Suggestions for Authors

Author Response

We sincerely appreciate your insightful feedback throughout the review process. Your comments provided us with the opportunity to significantly enhance the quality of our paper. We believe this article will be able to present research results in Electronics.

Reviewer 2 Report

Comments and Suggestions for Authors

The authors have answered most of my questions, I have just the last one suggestion.

For the rigor and completeness of the paper, please supplement the performance analysis under different SNR scenarios in the experiment.

Author Response

Thank you for your valuable feedback. Through your review, we believe this article will be able to present research results in Electronics. Furthermore, the quality of the article has been enhanced through your feedback.
